# Molecular dynamics simulations of a multicellular model with cell-cell interactions and Hippo signaling pathway

**Toshihito Umegaki**[1☯]*, **Hisashi Moriizumi**[2☯], **Fumiko Ogushi**[3], **Mutsuhiro Takekawa**[2], **Takashi Suzuki**[1]

**1** The Center for Mathematical Modeling and Data Science, Osaka University, Osaka, Japan, **2** The Institute of Medical Science, The University of Tokyo, Tokyo, Japan, **3** Faculty of Mathematical Informatics, Meiji Gakuin University, Tokyo, Japan

☯ These authors contributed equally to this work.
* umegaki@sigmath.es.osaka-u.ac.jp

**Data Availability Statement:** All relevant data are within the manuscript and its Supporting information files.

## Abstract

The transcriptional coactivator Yes-associated protein (YAP)/transcriptional co-activator with PDZ binding motif (TAZ) induces cell proliferation through nuclear localization at low cell density. Conversely, at extremely high cell density, the Hippo pathway, which regulates YAP/TAZ, is activated. This activation leads to the translocation of YAP/TAZ into the cytoplasm, resulting in cell cycle arrest. Various cancer cells have several times more YAP/TAZ than normal cells. However, it is not entirely clear whether this several-fold increase in YAP/TAZ alone is sufficient to overcome proliferation inhibition (contact inhibition) under high-density conditions, thereby allowing continuous proliferation. In this study, we construct a three-dimensional (3D) mathematical model of cell proliferation incorporating the Hippo-YAP/TAZ pathway. Herein, a significant innovation in our approach is the introduction of a novel modeling component that inputs cell density, which reflects cell dynamics, into the Hippo pathway and enables the simulation of cell proliferation as the output response. We assume such 3D model with cell-cell interactions by solving reaction and molecular dynamics (MD) equations by applying adhesion and repulsive forces that act between cells and frictional forces acting on each cell. We assume Lennard-Jones (12-6) potential with a softcore character so that each cell secures its exclusive domain. We set cell cycles composed of mitotic and cell growth phases in which cells divide and grow under the influence of cell kinetics. We perform mathematical simulations at various YAP/TAZ levels to investigate the extent of YAP/TAZ increase required for sustained proliferation at high density. The results show that a twofold increase in YAP/TAZ levels of cancer cells was sufficient to evade cell cycle arrest compared to normal cells, enabling cells to continue proliferating even under high-density conditions. Finally, this mathematical model, which incorporates cell-cell interactions and the Hippo-YAP/TAZ pathway, may be applicable for evaluating cancer malignancy based on YAP/TAZ levels, developing drugs to suppress the abnormal proliferation of cancer cells, and determining appropriate drug dosages. The source codes are freely available.

**Funding:** The work was supported by the following grants: M.T. received funding from the Japan Science and Technology Agency (JST), CREST (Grant No. JPMJCR2022). The funder's website: https://projectdb.jst.go.jp/grant/JST-PROJECT-20348631/. H.M. received funding from the Japan Society for the Promotion of Science (JSPS) (Grant No. 24K18108). The funder's website: https://kaken.nii.ac.jp/grant/KAKENHI-PROJECT-24K18108. The funders had no role in study design, data collection and analysis, decision to publish, or preparation of the manuscript.

**Competing interests:** The authors have declared that no competing interests exist.

## Author summary

This study numerically clarifies multicellular spatiotemporal properties by solving MD and reaction equations for application to any reaction network. We applied the Hippo-YAP/TAZ signaling pathway as a reaction network model for normal and cancer tissues. Moreover, we quantitatively discuss the cell proliferation outcomes in normal and cancerous tissues by calculating time-dependent multicellular structures on the 3D model using the MD method. This approach paves the way for the mathematical exploration of patterns in physical therapies, such as drug administration.

## 1 Introduction

The Hippo pathway has recently attracted attention as a molecular target for anticancer drugs and regenerative medicine [1]. This unique pathway senses cell contacts and mechanical stimulation, supports organ size control and cell differentiation, and acts as a potent tumor suppressor gene [1]. The Hippo pathway is an evolutionarily conserved signaling cascade that controls organ size and tumorigenesis [2–4]. In mammals, the Hippo pathway involves two kinases: MST1/2 and LATS1/2. Under low-density conditions, MST1/2 and LATS1/2 remain inactive, allowing co-activators Yes-associated protein (YAP) and transcriptional co-activator with PDZ binding motif (TAZ) to interact with transcription factors, such as TEAD1-4, in the nucleus, thereby inducing the transcription of genes involved in cell proliferation and survival. In contrast, MST1/2 is activated first under high-density conditions, leading to the phosphorylation of LATS1/2. Activated LATS1/2 phosphorylates YAP/TAZ leads to its sequestration in the cytoplasm by binding to 14-3-3 proteins. Consequently, the YAP/TAZ-induced expression of cell proliferation-related genes is inhibited [5–10].

YAP and TAZ are major effectors of the Hippo pathway and are considered proto-oncoproteins involved in cancer tissue proliferation, stemness, metastasis, and drug resistance [11–13]. Indeed, in various cancers, increased YAP/TAZ levels have been observed because of mutations in MST1/2 and LATS1/2 as well as gene amplification of YAP1 and WWTR1, which encode YAP and TAZ [14–21]. However, the precise extent of YAP/TAZ protein upregulation in cancer tissues is several-fold higher than that in normal tissues [12, 22, 23], and measures to evade cell cycle arrest under high-density conditions and trigger sustained chronic proliferation [24] remains poorly understood.

Some methods use cell elasticity and adhesion [25], and potential forces with arguments such as exponential functions [26, 27] to model cell-cell interactions. Previous studies on cells using MD methods includes cases without proliferations [28], reactions and diffusions [29], dynamics of two-species particle systems [30], mitotic cone formations [31], multiparticle collisions in aqueous solutions [32], and 3D models with cell growths and divisions in the context of tumor invasions [25]. However, these studies often did not directly address the intricate molecular signaling pathways that regulate cell proliferation and their interactions with mechanical forces at the multicellular level. Additionally, previous models did not incorporate the dynamic feedback mechanisms of the Hippo pathway on cell proliferation and survival, especially under high cell density conditions.

This study investigated whether an increase in YAP/TAZ levels is sufficient to induce sustained chronic cancer proliferation. We conducted cell proliferation simulations using a 3D model that considered cell proliferation and cell-cell interactions to achieve such investigations based on reaction networks representing normal and cancerous tissues. We combined MD

models with a reaction network model of the Hippo pathway, which controls cell proliferation by inhibiting YAP/TAZ.

We mimicked normal and cancerous tissues using different initial concentrations of YAP/TAZ molecules. While previous studies used the Lennard-Jones (LJ) potential for modeling intermolecular forces in MD methods [33], we used the LJ potential for the adhesion and repulsion of multiple cells, which have softcore characteristics, avoiding sudden increase when the cells are close [34]. Our approach uniquely integrates the Hippo pathway signaling dynamics with cell-cell interactions, providing a more comprehensive understanding of the regulation of cell proliferation and tumorigenesis.

In this study, we calculated 3D spatiotemporal patterns of normal and cancerous tissues, including cell survival or cell cycle arrest. We added external factors, such as adhesion, repulsion, and friction, based on kinetic equations to describe the motion of an agent representing each cell. Here, we calculated multicellular time evolutions by solving reaction equations coupled with the MD method, where we built models to incorporate the factors of cell growth and division. Herein, a significant innovation in our approach is the introduction of a novel modeling component that inputs cell density, which reflects cell dynamics, into the Hippo pathway and enables the simulation of cell proliferation as the output response. In addition to the standard MD model, we introduce the cell cycle as a factor to reflect the dynamics of life in this model. We also solved the equations of motion of cells that undergo repeated growth and division and obtained temporal evolutions of the positions of multicellular cells. By integrating the molecular signaling of the Hippo pathway with cell mechanics and cell cycle dynamics, our study offers new insights into the mechanisms driving cancer proliferation and presents a newly and more dynamic model compared to previous research.

## 2 Materials and methods

We summarize the overall methodology for 3D multicellular simulations in Fig 1A. The green and orange boxes represent the simulations of cell dynamics and intra-cellular dynamics, respectively. The simulation of cell dynamics outputs the cell density ($\rho$), which is then transferred to the intra-cellular dynamics simulation. This simulation subsequently outputs the concentration of nuclear YAP/TAZ, which induces cell proliferation, back into the cell dynamics. In cancer tissues, YAP/TAZ concentrations are several times higher than in normal tissues,

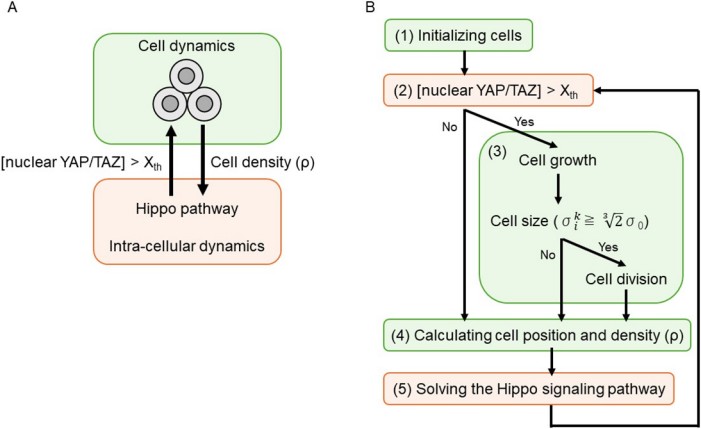

**Fig 1. Overall methodology for 3D multicellular simulation.** A: Interaction between cell dynamics and intra-cellular dynamics involving Hippo pathway. B: Flowchart depicting the five models.

leading to chronic proliferation. We replicated these experimental findings using a three-dimensional (3D) molecular dynamics (MD) model and reaction equation analysis. In the 3D MD model, cell positions and velocities were numerically analyzed using the Verlet method [35], a numerical technique employed in MD to integrate Newton's equations of motion, renowned for its accuracy and stability in long-term simulations, while assuming potential forces between the cells. Our novel approach within the model includes a sophisticated algorithm for cell division that is based on real-time changes in intracellular signaling pathways, particularly the Hippo-YAP/TAZ pathway. The reaction equations for the Hippo-YAP/TAZ pathway were numerically analyzed using the eigenvalue problem method. Further details of these five models are provided in Fig 1B and the following subsections.

## 2.1 Initializing cells

**Algorithm 1** Initializing cells and 3D multicellular simulation

```
 1: (1) Initializing cells:
```
**Input:** cell cycle: $T_{CC}$=20h, duration of mitosis phase: $T_M$=1h, total simulation time $t_L=L \times dt \cong 10.4 T_{CC} = 208$ h, $dt = 3.75$s, $L$=200000.

**Input:** $\gamma = \sqrt[3]{2}-1$, $\zeta = \sqrt[6]{2-\alpha_{LJ}(1-\lambda)^2}$,

**Input:** $\boldsymbol{e}_r={}^t(\sin\phi\cos\theta,\ \sin\phi\sin\theta,\ \cos\phi)$, $\theta$=random $[0, 2\pi)$, $\phi$=random $[0, \pi)$,

**Input:** $\sigma_0' = 5\mu$m, $m_0$=2 ng, $\sigma_0 = \sigma_0'/\zeta$, $d\sigma = (\sqrt[3]{2}-1)dt\sigma_0/T_{CC}$,

**Input:** $X_0$=0.15 [$\mu$M]

```
 2: N⁰=2
 3: if the tissue is normal then
```
```
 4:    Xᵢ⁰=ᵗ(0.05,0.2,0)[µM], i=1, 2
 5: else {the tissue is cancer}
 6:    Xᵢ⁰=ᵗ(0.1,0.4,0)[µM], i=1, 2
 7: end if
 8: αᵢ⁰ = 0, σᵢ⁰ = σ₀, rᵢ⁰ = ∓γσ₀'eᵣ, vᵢ⁰ = ∓(1−γ)σ₀'eᵣ/Tₘ, i=1, 2,
 9: k ← 1
10: while 1 ≤ k ≤ L do
11:    t = kdt
12:    (2) Discrimination of whether cell growth or cell cycle arrest
13:      In: αᵢᵏ, σᵢᵏ, X₂ᵢᵏ, Out: αᵢᵏ, σᵢᵏ, cᵢᵏ, i=1, ···, Nᵏ⁻¹.
14:    (3) Discrimination of whether cells division or nondivision
15:      In: Nᵏ⁻¹, σᵢᵏ, αᵢᵏ, rᵢᵏ, vᵢᵏ, X₁ᵢᵏ, ρᵢ, i=1, ···, Nᵏ⁻¹, l=1,2,3.
16:      Out: Nᵏ, σⱼᵏ, αⱼᵏ, rⱼᵏ, vⱼᵏ, X₁ⱼᵏ, ρⱼ, j=1, ···, Nᵏ, l=1,2,3.
17:    (4) Solving multicellular kinetic equations
18:      In: rⱼᵏ, vⱼᵏ, Out: rⱼᵏ, vⱼᵏ, j=1, ···, Nᵏ.
19:    (5) Solving reaction equations of each cell
20:      In: X₁ⱼᵏ, rⱼᵏ, Out: X₁ⱼᵏ, ρⱼᵏ, j=1, ···, Nᵏ, l=1,2,3.
21:    k ← k + 1
22: end while 1 ≤ k ≤ L
```
**Output:** $X_i^k$, $\boldsymbol{r}_i^k$, $\sigma_i'^k$, $c_i^k$

We assume that the two cells have initial positions $\boldsymbol{r}_1^0$ and $\boldsymbol{r}_2^0$ and velocities $\boldsymbol{v}_1^0$ and $\boldsymbol{v}_2^0$ with directions of the azimuthal angle in the xy-plane from the x-axis; $\theta$, and the polar angle from the positive z-axis; $\phi$ according to the pseudocode of Algorithm 1. where $\boldsymbol{r}_i^k$ and $\boldsymbol{v}_i^k \in \mathbb{R}^3$ are center positions of $i$-th cells, $i$=1, $\cdots$, $N^k$, $k$ is natural number representing the time step in time evolution, where $k = 0, \cdots, L$. $L$ is the maximum number for $k$, and the time $t$ is given by $t = kdt$. $N^k$ is the total number of cells at time step $k$; the initial value $N^0$ is set to 2 at time $t$=0, $\gamma$ represents the dimensional ratio between the mother cell just before division and the daughter cells just after division, calculated by subtracting one from the cube root of two, such that the

volume of one mother cell is equal to the sum of the volumes of the two daughter cells; $\gamma = (\sqrt[3]{2}-1)$. This assumption is based on previous researches [36, 37] that determined cell division is regulated by whether the cell size exceeds a certain threshold or not.

$^t(x, y, z)$ is a vector with components $x$, $y$, $z$ in the Cartesian coordinate system, $e_r$ is the radial unit vector in the polar coordinate system. $\theta$ and $\phi$ are uniform random numbers $[0, 2\pi)$ and $[0, \pi)$, respectively; the directions of cell division are random [38], and the upper-right subscripts represent the discrete values at time $t=kdt$. Algorithm 1 describes the calculations of the entire cell-cell interaction model of elements (1)–(5); details of (2)–(5) are described in the following sections.

**2.1.1 Modeling of cancer and normal tissues.** To investigate whether the increased levels of YAP/TAZ in cancer tissues led to sustained chronic proliferation, we used two models representing normal and cancerous tissues. In this simulation, we defined the variables $X_0$ and initial values of $X_{1,i}$–$X_{3,i}$ as presented in Table 1. The parameters are set as follows:

1. As an input signal transmitted from surrounding cells to $i$-th cells with the cell density $\rho_i=1$, $X_0$ is set to 0.15 [$\mu$M], and is an arbitrary value independent of normal or cancer tissues.

2. YAP/TAZ concentrations $X_{1i}$ and $X_{2i}$ are of the order of 0.01–1[$\mu$M] from Table S1 of [39]. At the initial low cell density, the YAP/TAZ predominantly localizes to the nucleus rather than the cytoplasm [5]. Therefore, the initial concentration ratio of cytoplasm and nucleus $X_{1i}^0/X_{2i}^0$ is set to 1/4, that is, $X_{1i}^0 = 0.05[\mu M]$, and $X_{2i}^0 = 0.02[\mu M]$ for normal tissues.

3. $X_{1i}^0$ and $X_{2i}^0$ are set to be several times higher in cancer tissues compared to normal tissues since cancer tissues exhibit YAP/TAZ levels several times higher than those in normal tissues [12, 22, 23]. Therefore, in this simulation, we consider the YAP/TAZ concentration in cancer tissues to be twice that of normal tissues, that is, the ratio of $X_{1i}^0$ or $X_{2i}^0$ between normal and cancer tissues is set to 1/2. Thus, the initial concentrations of normal and cancer tissues $(X_{1i}^0, X_{2i}^0)$ are set to (0.05, 0.2) and (0.1, 0.4) [$\mu$M], respectively.

4. The reaction coefficients $a_1$ and $a_2$ are of the order of $10^{-4}$- 1 [/s] from Table S2 of [39]. YAP/TAZ is localized in the nucleus compared to the cytoplasm [5]. Thus, $a_1/a_2$ is set to 100, that is, $a_1=5\times10^{-2}$[/s], $a_2=5\times10^{-4}$[/s].

**Table 1. Parameters for the reaction equations [5, 22, 39].**

| Parameter | Definition | Value [unit] |
|---|---|---|
| Concentration | $X_0$: input transmitted into a cell when $\rho_i=1$ | 0.15 [$\mu$M] |
| Initial concentration | $X_{1i}^0$: YAP/TAZ cytoplasm | |
| | in normal tissues | 0.05 [$\mu$M] |
| | in cancer tissues | 0.1 [$\mu$M] |
| | $X_{2i}^0$: YAP/TAZ nucleus | |
| | in normal tissues | 0.2 [$\mu$M] |
| | in cancer tissues | 0.4 [$\mu$M] |
| | $X_{3i}^0$: P-YAP/TAZ cytoplasm | 0 [$\mu$M] |
| Reaction rate | $a_1$: YAP/TAZ cytoplasm $\rightarrow$ nucleus | $5\times10^{-2}$[/s] |
| | $a_2$: YAP/TAZ nucleus $\rightarrow$ cytoplasm | $5\times10^{-4}$ [/s] |
| | $b_1$: YAP/TAZ cytoplasm $\rightarrow$ P-YAP/TAZ | 1 [/($\mu$M· s)] |
| | $a_3$: P-YAP/TAZ $\rightarrow$ YAP/TAZ cytoplasm | $1\times10^{-4}$ [/s] |

5. The reaction coefficients $b_1$ and $a_3$ are of the order of $10^{-4}$- 1[/s] from Table S2 of [39]. When cells are tightly packed with $\rho_i \cong 1$, phosphorylated YAP/TAZ is almost tethered in cytoplasmic YAP/TAZ compared to nuclear YAP/TAZ [5]. Thus, $b_1 X_0$ is set to about 1000 times $a_3$, $b_1 X_0$=0.1[/s], $a_3$=$10^{-4}$[/s], and $b_1$=0.1[/s]/0.1[$\mu$M]=1[/$\mu$Ms].

## 2.2 Discriminate between cell growth and cell cycle arrest

Depending on whether the Nuclear YAP/TAZ of the $i$-th cell, $X_{2i}^k$, was larger or smaller than the concentration threshold $X_{th}$, we determined whether the cells experience growth or cell cycle arrest. We express these states using four cell colors defined by the cell color scheme in Fig 2; cell colors of orange and red correspond to the mitotic [M ($\alpha_i < T_M$)] and cell growth [G1 ($\alpha_i > T_M$)] phases of growing cells; where $\alpha_i$ is the age of $i$-th cell, that is, the time immediately after division, and cell colors of cyan and blue correspond to the M and G1 phases of the cell cycle arrest, respectively. $c_i$ is color index of $i$-th cell, which assigned for the colors of cyan, orange, blue and red to 0, 1, 2, and 3, respectively. During the aging of $i$-th cell from 0 to $T_{CC}$, the cells grow from $\sigma_i=\sigma_0$ to $\sigma_i = \sqrt[3]{2}\sigma_0$ if $X_{2i} > X_{th}$ where $T_{CC} \equiv T_M + T_{G1}$ is the cell cycle, $T_M$ and $T_{G1}$ are the periods of the M and G1 phases, respectively. The cell radius in the M phase is $\sigma_0'$ immediately after cell division, and the sizes of the two daughter cells are within the range of one mother cell, as indicated by the red dashed circle. The green two-way arrows on the left and right sides of Fig 2 indicate that the red and orange cells in the growing state and the blue and cyan cells in the cell-cycle-arrest state alternate. The $i$th cell is discriminated as belonging to the growing state (down green arrow) or cell cycle arrest (up green arrow) if $X_{2i}$ becomes greater or less than $X_{th}$, respectively. In the M phase, the cells first divide into two daughter cells at a distance of $\pm\gamma\sigma_0'$ from the central position of the mother cell. From age 0 to $T_M$, the cells start growing at a growth rate $d\sigma/dt=\gamma\sigma_0/T_{CC}$ and the multicellular structures are rearranged due to the equilibrium of forces acting among the cells. At ages $T_M$ and above in the G1 phase, each cell grows and moves independently. With cell cycle arrest, $\alpha_i$ remains constant, while in cell growth, the age of the $i$-th cell, $\alpha_i$, increases. When $\alpha_i$ reaches time $T_{CC}$, the cell divides. If the cell remains in a red or orange state over time $T_{CC}$, it's defined as one cycle.

We define the cell growth processes for each $dt$ progress at time $t=kdt$, $k=1, \cdots, L$, if $\sigma_0 < \sigma_i^k < \sqrt[3]{2}\sigma_0$ and $X_{2i} > X_{th}$ as follows:

$$\alpha_i \leftarrow \alpha_i + dt, i = 1, \cdots, N, \tag{1a}$$

$$\sigma_i \leftarrow \sigma_i + d\sigma, i = 1, \cdots, N. \tag{1b}$$

## 2.3 Discriminate between cell division and nondivision

We assume cell division if $i$-th cell becomes $\sigma_i^k \geq \sqrt[3]{2}\sigma_0$ at time $t=kdt$, $k = 1, \cdots, L$, which assumption is based on previous studies [36, 37] that have determined cell division is regulated by whether the cell size exceeds a certain threshold or not. The number, position, and velocity of cells are updated as follows:

$$N \leftarrow N + 1, \tag{2a}$$

$$\sigma_i \leftarrow \sigma_0, \tag{2b}$$

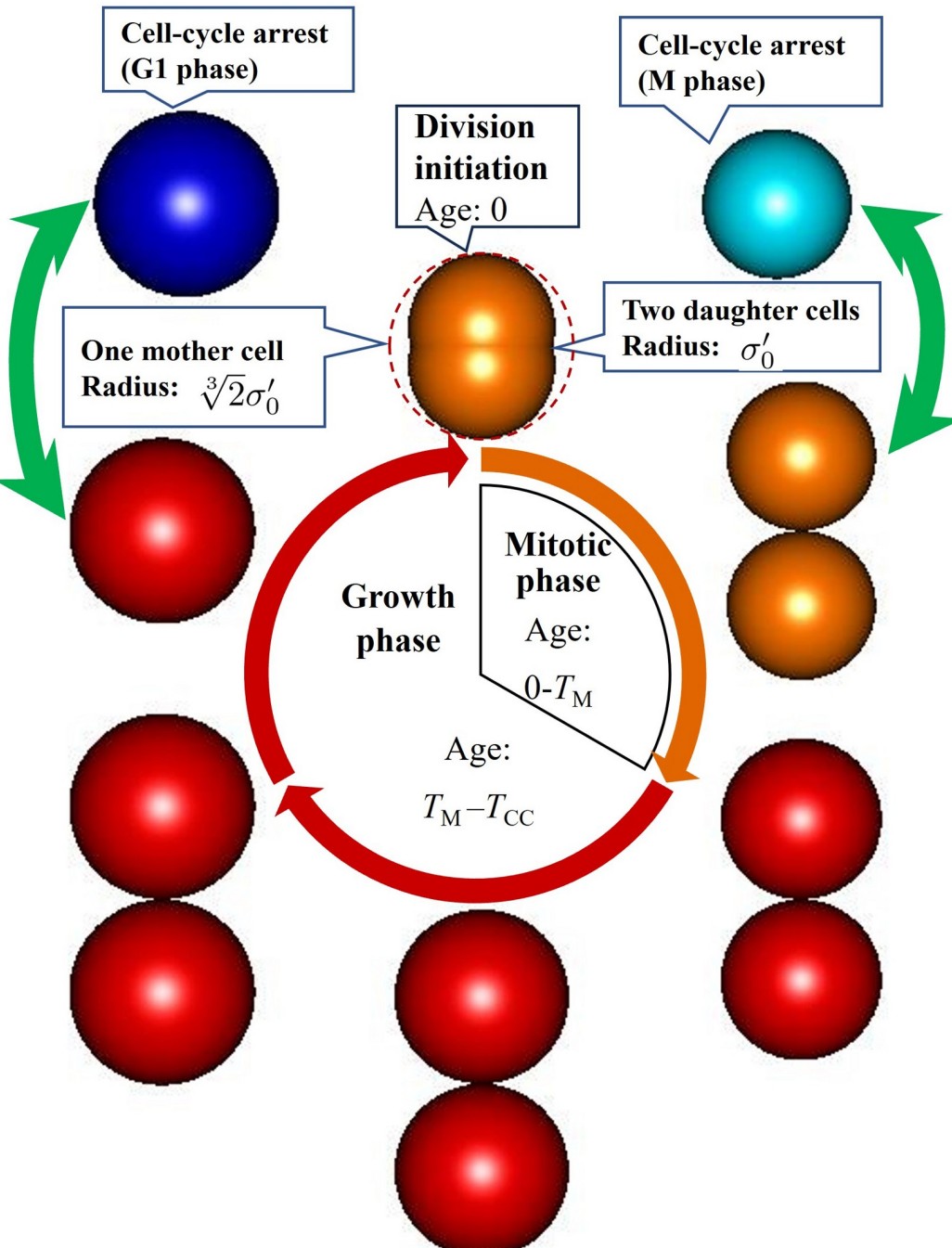

**Fig 2. Cell cycle phases and cell color scheme definitions.** Growing cells in M and G1 phases are shown as orange and red spheres, respectively; cell-cycle-arrest cells in M and G1 phases are shown as cyan and blue spheres, respectively. The cell radius in the M phase is $\sigma'_0$ immediately after cell division, and the two daughter cell size is within the range of the mother cell indicated by the red dashed circle. The green two-way arrows on the left and right indicate that cells in the growing and cell-cycle-arrest states alternate. The $i$-th cells are discriminated in growing state (down green arrow) or cell-cycle-arrest (up green arrow) if $X_{2i}$ becomes more or less than $X_{th}$, respectively.

$$\sigma_N \leftarrow \sigma_0, \tag{2c}$$

$$\alpha_i \leftarrow 0, \tag{2d}$$

$$\alpha_N \leftarrow 0, \tag{2e}$$

$$\boldsymbol{r}_i \leftarrow \boldsymbol{r}_i - \gamma\sigma'_0\boldsymbol{e}_r, \tag{2f}$$

$$\boldsymbol{r}_N \leftarrow \boldsymbol{r}_i + \gamma\sigma'_0\boldsymbol{e}_r, \tag{2g}$$

$$\boldsymbol{v}_i \leftarrow -(1-\gamma)\sigma'_0\boldsymbol{e}_r/T_{\mathrm{M}}, \tag{2h}$$

$$\boldsymbol{v}_N \leftarrow +(1-\gamma)\sigma'_0\boldsymbol{e}_r/T_{\mathrm{M}}, \tag{2i}$$

$$X_{li} \leftarrow X_{li}, \tag{2j}$$

$$X_{lN} \leftarrow X_{li}, \tag{2k}$$

$$c_i \leftarrow 1, \tag{2l}$$

$$c_N \leftarrow 1. \tag{2m}$$

If $\sigma_i^k < \sqrt[3]{2}\sigma_0$, $N^k$, $\alpha_i^k$, $\boldsymbol{r}_i^k$ and $\boldsymbol{v}_i^k$, inherit the values at previous time $t=(k-1)dt$.

## 2.4 Solve multicellular kinetic equations using MD method

By solving the kinetic differential equations using the Verlet method S2 Appendix, we calculated the time evolutions of the cell positions $\boldsymbol{r}_i^k$ and velocities $\boldsymbol{v}_i^k$, $i = 1, \cdots, N^k$, $k = 0, \cdots, L$.

## 2.5 Solve reaction equations of each cell in the multicellular system

In this section, we construct a model of reaction network of the Hippo-YAP/TAZ signaling pathway [40] in Fig 3 an example. The corresponding reaction equations are given by Eqs (3a), (3b) and (3c). $X_{1i}^0$, $X_{2i}^0$, and $X_{3i}^0$ are the concentrations of YAP/TAZ cytoplasm, YAP/TAZ nucleus, and P-YAP/TAZ cytoplasm, respectively. Translocation, phosphorylation, dephosphorylation, and phosphate are indicated by orange, green, and blue arrow lines and circled P marks, respectively, in Fig 3. Accordingly, the reaction equations for the molar concentrations $X_{li}$ of $l$-th molecule in $i$-th cell are given as follows:

$$\frac{dX_{1i}}{dt} = -a_1X_{1i} + a_2X_{2i} + a_3X_{3i} - b_1\rho_iX_0X_{1i}, \tag{3a}$$

$$\frac{dX_{2i}}{dt} = +a_1X_{1i} - a_2X_{2i}, \tag{3b}$$

$$\frac{dX_{3i}}{dt} = -a_3X_{3i} + b_1\rho_iX_0X_{1i}, \tag{3c}$$

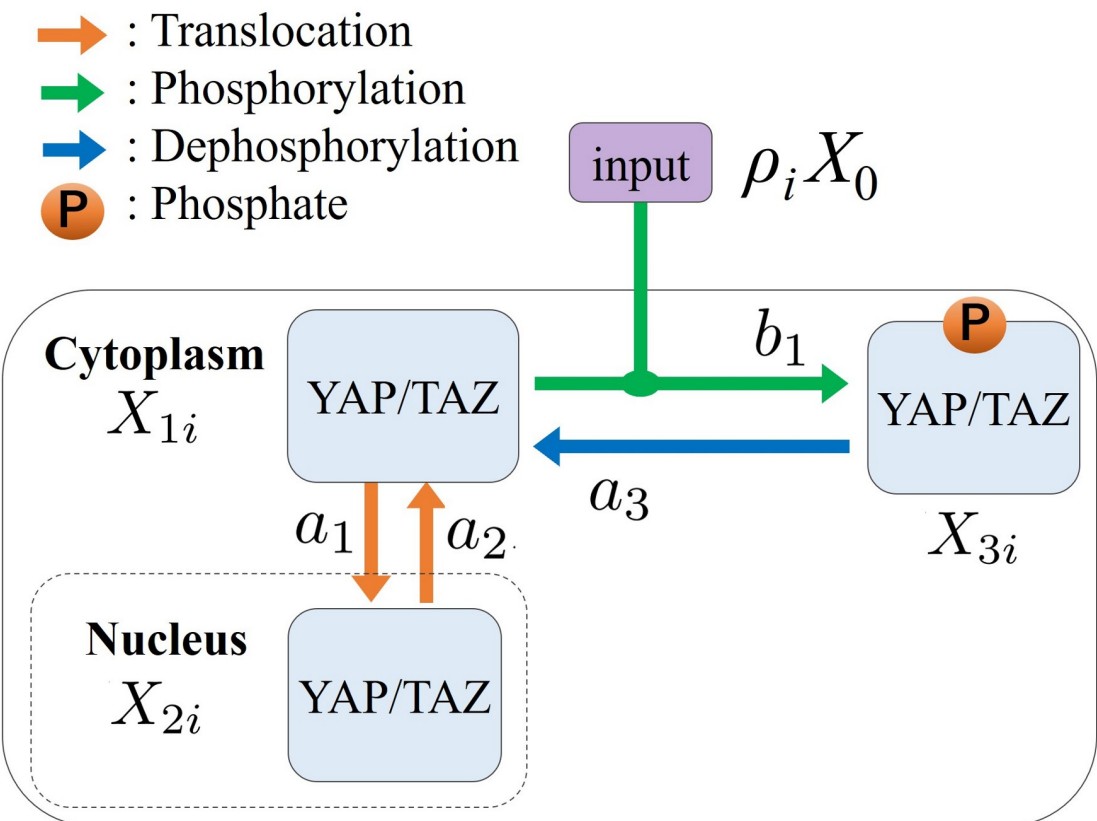

**Fig 3. Reaction network model of the Hippo-YAP/TAZ signaling pathway [40].** The corresponding reaction equations are Eqs (3a), (3b) and (3c). $X_{1i}^0$, $X_{2i}^0$, and $X_{3i}^0$ are concentrations of YAP/TAZ cytoplasm, YAP/TAZ nucleus, and P-YAP/TAZ cytoplasm, respectively. Orange, green, and blue arrowed lines and circled P marks indicate translocation, phosphorylation, dephosphorylation, and phosphate, respectively.

The reaction equation parameters are presented in Table 1 [5, 22, 39]. The derivation process of the model and the simulations are explained in S1 Appendix. We can also diagonalize the right-hand sides of Eqs (3a), (3b) and (3c), which we discuss later in Discussion and S3 Appendix using the eigenvalue problem method. We calculate the packing fraction $\rho_i$ of the $i$-th cell, which is defined as the sum of exponential functions of the distance between the $i$-th cell and other multicellular distances formulated in S4 Appendix, where the cell position $r_i$, $i=1, \cdots, N$ obtained using MD calculations. We can solve these reaction equations in conjunction with the MD equations when calculating $\rho_i$ in S4 Appendix.

## 3 Results

This section calculates the time evolution of molar concentrations and 3D multicellular spatio-temporal properties. We set the initial cell radius to $\sigma_0' = 5$ $\mu$m, with $\sigma_0 = \sigma_0'/\zeta = 4.53$ $\mu$m; The parameters are presented in Table B1 of S2 Appendix, where $\zeta$ is the ratio of cell radius to the $\sigma_i$ of $i$-th cell; $\zeta = \sqrt[6]{2 - \alpha_{\mathrm{LJ}}(1 - \lambda)^2}$, where $\alpha_{\mathrm{LJ}}$ is parameter of the LJ potential, $\lambda$ is a coupling parameter. For normal and cancerous tissues, we set the initial molar concentrations to $(X_{1i}^0, X_{2i}^0, X_{3i}^0) = (0.05, 0.2, 0)$ and $(0.1, 0.4, 0)[\mu$M], respectively, as listed in Table 1 [22]. We set cell

growth to occur when the nuclear levels of YAP/TAZ ($X_{2i}$) reach or exceed a specific threshold value $X_{th}$, where $X_{th}$ was set to 0.012 $\mu$M experimentally.

## 3.1 Time evolutions of molecular concentrations

We find solutions to the simultaneous first-order differential equations, Eqs (3a), (3b) and (3c), to calculate the time evolutions of the multicellular cells with the parameters in Table B1 of S2 Appendix using the eigen-value problem (EVP) formulations in S3 Appendix. The Time evolution of nuclear YAP/TAZ concentrations in normal and cancer tissues are shown in Fig 4A and 4B, respectively. The red, green, and blue solid lines represent maximum, mean, and minimum of $X_{2i}$ for $i$=1$\cdots N$ (total number of multicellular cells) defined as follows:

$$X_l^{\text{mean}} = \frac{1}{N}\sum_{i=1}^{N}X_{li}, \qquad (4a)$$

$$X_l^{\text{max}} = \max_{i=1,\cdots,N}(X_{li}), \qquad (4b)$$

$$X_l^{\text{min}} = \min_{i=1,\cdots,N}(X_{li}), \qquad (4c)$$

where $X_2^{\text{max}}$, $X_2^{\text{mean}}$, and $X_2^{\text{min}}$ are the mean, maximum, and minimum molecular concentrations $X_{li}$, $l$=1, 2, 3, respectively. The black dashed lines represent the threshold $X_{th}$. We discriminate between cell proliferation and growth arrest depending on whether $X_{2i}$ exceeds $X_{th}$. If $X_2^{\text{min}}$ falls below $X_{th}$, the cells experience growth arrest. For normal tissues, we observe cell cycle arrests in addition to cell proliferation, especially 100 tens of hours later in the time evolution in Fig 4A, where $X_2^{\text{min}}$ is lower than $X_{th}$. In contrast, for cancer tissues, $X_2^{\text{min}}$ do not fall below $X_{th}$ until around 200 h, as shown in the time evolution in Fig 4B.

## 3.2 Time evolutions of 3D real-images

We calculated 3D real-images shown in Fig 5 of (A, B) normal and (C, D) cancer tissues by solving the multicellular equations of motions in S2 Appendix using the Verlet method using recursion formulas Eq. (1.5) in [41] and coupled with the reaction equations Eqs (3a), (3b) and (3c). Panels (A, C) and (B, D) show the perspective view and cross-section at $x$=0 of the multi-cells at $t$= 208 hours (10.4 cell cycles) immediately after a cell dividing, respectively. The scale of these images is in [$\mu$m]. Colors of the spherical cells are orange and red for cell growth in the M and G1 phases and cyan and blue for cell cycle arrest in the M and G1 phases, respectively. We show videos of 3D perspective views for normal and cancer tissues over 0–208 h

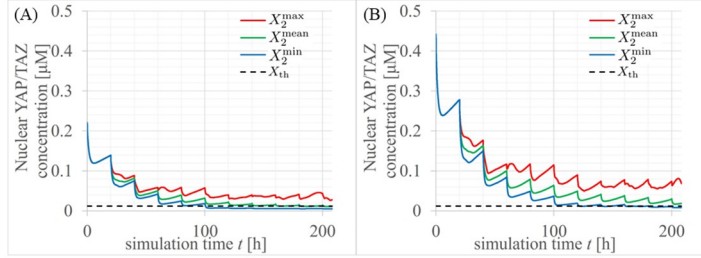

**Fig 4. Time evolutions of nuclear YAP/TAZ concentrations in (A) normal and (B) cancer tissues.** Red, green, and blue solid lines represent maximum, mean, and minimum of $X_{2i}$ for $i$=1, $\cdots$, $N$ (total number of multicellular cells) defined as $X_2^{\text{max}}$, $X_2^{\text{mean}}$, and $X_2^{\text{min}}$ in Eq (4), respectively. The black dashed lines represent the threshold $X_{th}$.

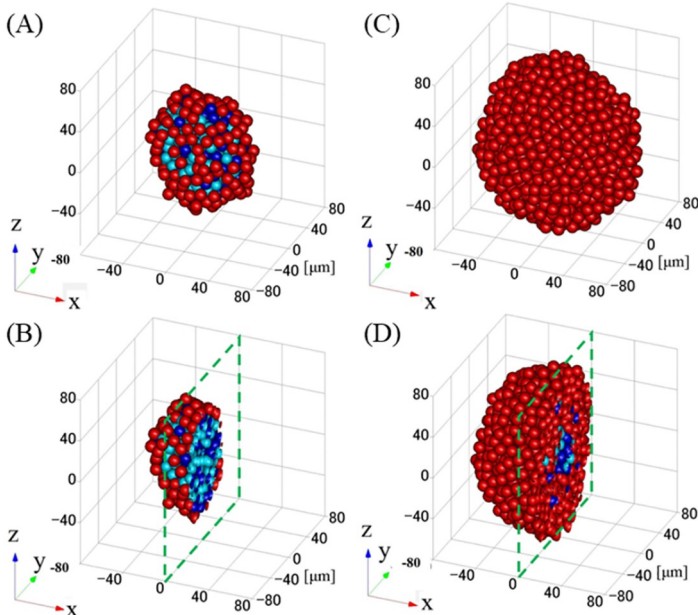

**Fig 5. 3D real-images of (A, B) normal and (C, D) cancer tissues.** Panels (A, C) and (B, D) display perspective views and cross-sections at *x*=0 of the multicellular structures at *t*= 208 h (10.4 cell cycles), immediately after cell division, respectively. The scale of these images is in [*μm*]. Orange and red represent cell-growth, while cyan and blue indicate cell-cycle-arrest in M and G1 phases, respectively. The time point 208 hours (10.4 cell cycles) marks when the cell tissues have reached sufficiently growth. Dashed green squares denote the cross-sections at *x*=0.

immediately after cell division, as shown in the links S1 and S2 Videos, respectively, and cross-sections of 3D moving images for normal and cancer tissues, as shown in S3 and S4 Videos. The cross-sections show cuts at *x*=0 in the perspective views. To quantitatively evaluate the cell proliferation outcomes in normal and cancerous tissues, we calculated the time-dependent changes in the total number of multicellular structures, as shown in Fig 6 (0–208 hours). Exponential cell growth is suppressed in normal tissues, particularly in the center of tissues. However, in cancer tissues, despite having YAP/TAZ levels only twice as high as those in normal tissues, cancer tissues demonstrated exponential cell proliferation.

## 4 Discussion

We explored the effects of increased YAP/TAZ levels on cell proliferation in cancer cells. YAP/TAZ play a role in the cell nucleus and are restrained by the activated Hippo pathway under high-density conditions, causing them to be retained in the cytoplasm and ultimately stopping cell growth. Many cancer cells have significantly higher YAP/TAZ expression compared to normal cells [12, 22, 23]. However, it remains uncertain whether this increase in YAP/TAZ expression can override the inhibitory effect of the Hippo pathway and maintain cell growth. We developed a computational model using MD simulations to investigate this type of chronic proliferation maintenance. This model mimics cell interactions driven by external factors such as adhesion, repulsion, and friction.

    This study focuses on a model that integrates cell dynamics, which affect cell density, and intra-cellular dynamics, which influence YAP/TAZ levels. Our goal was to understand how increased YAP/TAZ levels in cancer cells affect their proliferation under high-density conditions. Using this model, we monitored the changes in nuclear YAP levels ($X_{2i}$) over time in

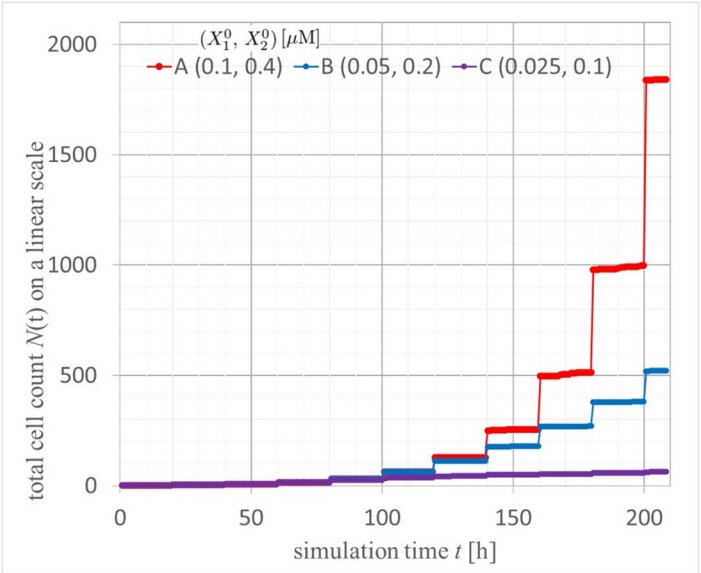

**Fig 6. Time evolutions of the total number of multi-cells of cancer and normal tissues and an exception condition with $(X_1^0, X_2^0) = (0.1, 0.4)$, (0.05, 0.2), and (0.025, 0.1), respectively.**

normal tissues. After 100 h, the minimum $X_{2i}$ fell below the threshold ($X_{th}$) required for nuclear YAP levels to support proliferation.

Additionally, in normal tissues, we observed exponential cell proliferation for 100 tens of hours, followed by cell cycle arrest in the central region of the cell population (Fig 5A and 5B), reducing exponential growth (Fig 6). In contrast, in cancer tissues, despite a twofold increase in YAP/TAZ expression compared to normal tissues, cell cycle arrest was rarely observed in the central region (high-density conditions) of the cell population (Fig 5C and 5D). Instead, exponential cell proliferation continued for up to 200 h (Fig 6). Fig 7 shows time evolutions of the total number of multi-cells of cancer and normal tissues with $(X_1^0, X_2^0) = (0.1, 0.4)$, (0.05, 0.2), and an exponential function with a time rounding-up parameter: $\exp\lceil t/20h \rceil$. From this semi-log plot, it is quantitatively confirmed that the red curve for cancer cells almost matches the exponential function of the orange dashed curve up to 200 hours, indicating that exponential cell growth continues up to that point. The basis for choosing this time point to draw conclusions is the nature of cancer cells, specifically as a model demonstrating exponential cell growth.

As shown in Fig 4B, in cancer tissues, $X_{2i}$ rarely fell below $X_{th}$ even after 200 h. Thus, a twofold increase in YAP/TAZ levels was sufficient to override the density-dependent growth inhibition effect and sustain exponential proliferation, thereby increasing cancer malignancy.

This model, integrating cell dynamics and intra-cellular dynamics, provides a comprehensive understanding of the influence of both cell density and YAP/TAZ concentration on proliferation, contributing to our understanding of the proliferative characteristics of normal and cancer cells.

From the comparison between Fig 5A–5C and 5D, the following computational and biological observations can be made despite only a twofold increase in YAP/TAZ levels:

1. **Tissue size increase:** The data indicates an increase in overall tissue size, suggesting a dose-dependent effect of YAP/TAZ on tissue growth.

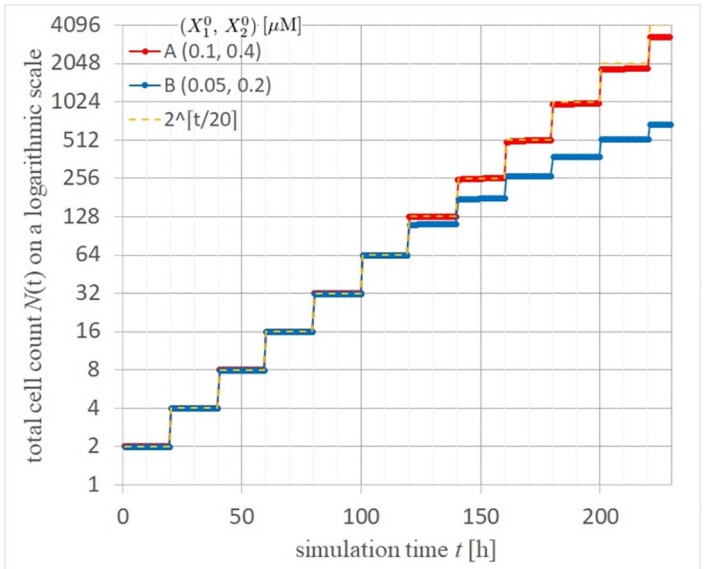

**Fig 7. Time evolutions of the total number of multi-cells of cancer and normal tissues with $(X_1^0, X_2^0) = (0.1, 0.4)$, $(0.05, 0.2)$, and an exponential function with a time rounding-up parameter: $\text{Exp}\lceil t/20h \rceil$.**

2. **Cell cycle arrest in normal vs. cancer cells:** As seen in Fig 5A and 5C, in normal cells, the cell cycle is arrested even in the peripheral regions of the tissue. In contrast, cancer cells in the same peripheral regions continue to proliferate, demonstrating dysregulation in the normal cell cycle control mechanisms.

3. **Induction of cell cycle arrest in tissue cross-sections:** The cross-section of Fig 5B reveals that in normal cells, cell cycle arrest is induced as cells transition from the outer to the inner layers. However, in cancer cells, cell cycle arrest is predominantly induced in the central parts of the tissue only, highlighting a spatially restricted response to growth inhibition signals in cancerous conditions.

Moreover, Fig 6 quantitatively demonstrates that cells typically double with each division, leading to exponential growth if unchecked. In normal tissues, this exponential increase is significantly suppressed. Remarkably, cancer cells with merely a twofold increase in YAP/TAZ levels exhibit almost no growth inhibition and continue to increase exponentially. This suggests that cancer cells with elevated YAP/TAZ levels bypass typical proliferative checks, underscoring the pivotal role of YAP/TAZ in regulating cellular proliferation dynamics in oncogenic contexts. The ratio of $X_2$ between cancer and normal cells (C/N_X2mean) exhibits a temporal evolution in Fig 8. Here, by setting the initial ratio of $X_2(cancer)/X_2(normal)$ to 2, this ratio remains around 2 for up to 100 hours. Interestingly, after 100 hours, regions where the ratio approaches threefold emerge. This nonlinear region mediated by multicellularity, concerning nuclear YAP/TAZ concentration, can be seen as the reason why cancer cells continue to proliferate without showing cell cycle arrest.

Reports have linked higher YAP protein levels to a poorer prognosis in patients with cancer [11, 13, 21]. Currently, clinical trials are underway to investigate drugs, such as VT3989, that inhibit the nuclear activity of YAP/TAZ. However, excessive doses of these drugs hinder the proliferation of normal tissues. Using the proposed 3D mathematical model, calculating the appropriate dose of VT3989 based on YAP/TAZ levels may become possible.

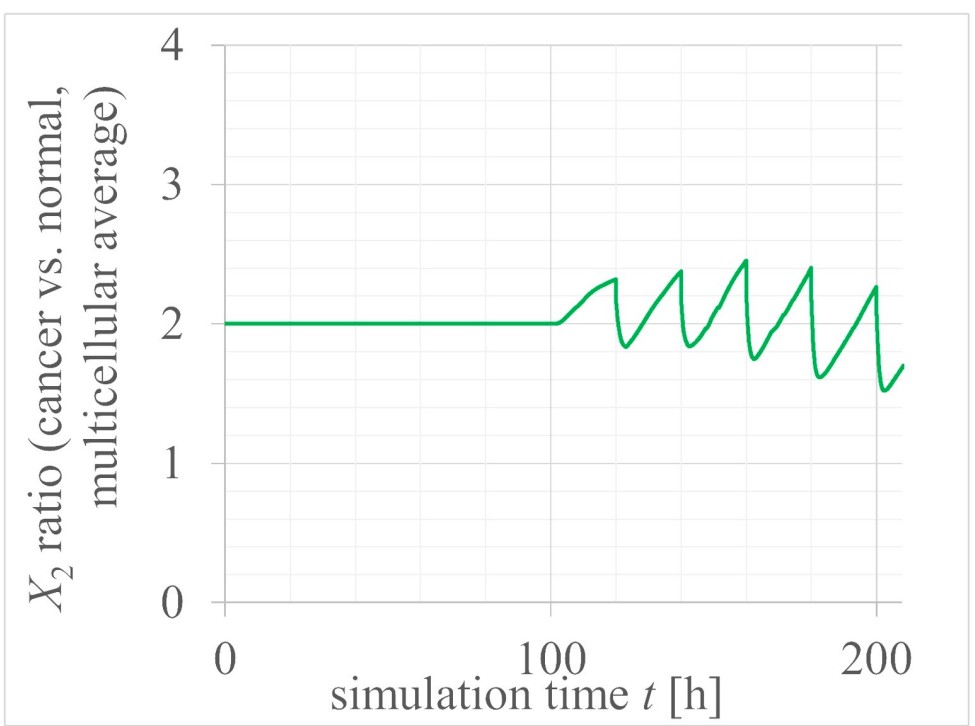

**Fig 8. Time evolution of the ratio of $X_2$ between cancer and normal cells.**

To investigate the influence of the packing fraction $\rho_i$ on concentrations, we define the solutions of the reaction equations, Eqs (3a), (3b) and (3c), for $\rho_i=\rho_0$ as $X_l^{\rho=\rho_0}$, where $\rho_0$ is a certain density. On Fig 9A and 9C, for normal and cancer tissues, we plot time evolutions of $X_2^{\mathrm{mean}}$, $X_2^{\rho=0.0}$, $X_2^{\rho=0.5}$, $X_2^{\rho=1.0}$, and $X_2^{\rho=1.5}$ as a green solid line and brown, orange, cyan, purple dashed lines, respectively. On Fig 9B and 9D, we also plot time evolutions of $X_3^{\mathrm{mean}}$, $X_3^{\rho=0.0}$, $X_3^{\rho=0.5}$, $X_3^{\rho=1.0}$, and $X_3^{\rho=1.5}$ as a green solid line and brown, orange, cyan, purple dashed lines, respectively. The time evolution of YAP/TAZ concentrations $X_2$ and $X_3$ falls within the range of $\rho=0$ to 1.5 in both normal and cancer tissues.

## 5 Conclusion

In this study, we performed 3D multicellular simulations combining MD methods with reaction network models of the Hippo pathway, which is known to regulate cell proliferation by inhibiting YAP/TAZ activity. Our approach effectively simulated cell division and growth, preventing cell overlap by assigning exclusive domains to each cell with LJ (12–6) potential with a softcore character, and integrating cell cycle dynamics, particularly the M and G1 phases. We introduced a novel modeling component that inputs cell density, reflecting cell dynamics, into the Hippo pathway simulation. This innovation allowed us to simulate cell proliferation as an output response, providing a more detailed and dynamic representation of how intracellular signaling modulates cell division decisions.

To investigate whether the elevated levels of YAP/TAZ observed in cancer tissues are sufficient to sustain chronic proliferation, we developed models representing time evolutions of normal and cancerous tissues. Our 3D simulations revealed that while normal tissues exhibited cell cycle arrest and suppression of exponential growth, cancer tissues with YAP/TAZ levels

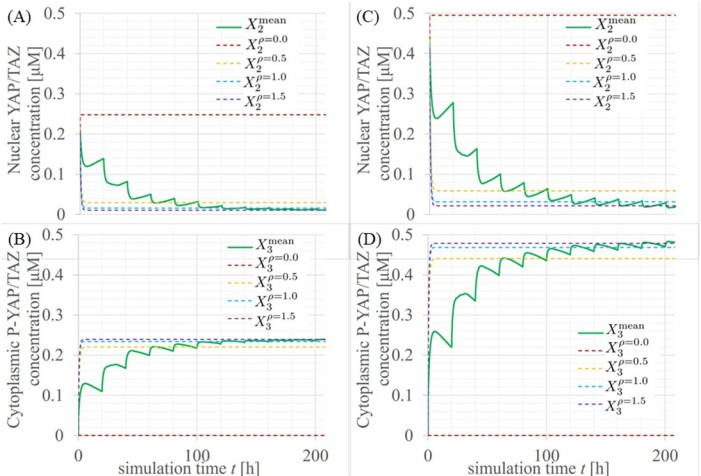

**Fig 9. Time evolutions of nuclear YAP/TAZ concentrations in (A, B) normal and (C, D) cancer tissues.** The green solid lines indicate mean values of (A, C) $X_{2i}$ and (B, D) $X_{3i}$ for $i=1\cdots N$ (total number of multicellular cells). The brown, orange, cyan, and purple dashed lines are solutions of the reaction equations Eqs (3a), (3b) and (3c) at cell packing fractions $\rho_i = 0.0$ to 1.5, respectively.

twice as high demonstrated continuous exponential proliferation. These findings suggest that increased YAP/TAZ alone can bypass contact inhibition under high-density conditions, contributing to unchecked proliferation in cancer cells. Our model's integration of cell dynamics and intra-cellular dynamics could position it as a tool for evaluating cancer malignancy, exploring therapeutic interventions targeting YAP/TAZ, and optimizing drug dosages.

## Supporting information

**S1 Video. Perspective 3D moving images of normal tissues over 0–208 h, scale unit [$\mu$m]; orange, red, and blue spherical cells represent M, G1, and cell-cycle-arrest phases, respectively.**
(MP4)

**S2 Video. Perspective 3D moving images of cancer tissues over 0–208 h, scale unit [$\mu$m]; orange, red, and blue spherical cells represent M, G1, and cell-cycle-arrest phases, respectively.**
(MP4)

**S3 Video. Cross-sections of 3D moving images of normal tissues over 0–208 h, scale unit [$\mu$m]; orange, red, and blue spherical cells represent M, G1, and cell-cycle-arrest phases, respectively.**
(MP4)

**S4 Video. Cross-sections of 3D moving images of cancer tissues over 0–208 h, scale unit [$\mu$m]; orange, red, and blue spherical cells represent M, G1, and cell-cycle-arrest phases, respectively.**
(MP4)

**S1 Appendix. Derivation and Simulation of the Computational Model.**
(PDF)

**S2 Appendix. Method to solve the kinetic equations using Verlet method.**
(PDF)

**S3 Appendix. Method to solve reaction equations using eigenvalue problem.**
(PDF)

**S4 Appendix. Packing fraction of each cells.**
(PDF)

## Author Contributions

**Conceptualization:** Toshihito Umegaki, Hisashi Moriizumi, Fumiko Ogushi, Mutsuhiro Takekawa, Takashi Suzuki.

**Data curation:** Toshihito Umegaki, Hisashi Moriizumi, Fumiko Ogushi.

**Formal analysis:** Toshihito Umegaki, Hisashi Moriizumi, Fumiko Ogushi.

**Funding acquisition:** Toshihito Umegaki, Mutsuhiro Takekawa, Takashi Suzuki.

**Investigation:** Toshihito Umegaki, Hisashi Moriizumi.

**Methodology:** Toshihito Umegaki, Hisashi Moriizumi, Fumiko Ogushi.

**Project administration:** Mutsuhiro Takekawa, Takashi Suzuki.

**Resources:** Toshihito Umegaki, Hisashi Moriizumi.

**Software:** Toshihito Umegaki.

**Supervision:** Mutsuhiro Takekawa, Takashi Suzuki.

**Validation:** Toshihito Umegaki, Hisashi Moriizumi.

**Visualization:** Toshihito Umegaki, Hisashi Moriizumi.

**Writing – original draft:** Toshihito Umegaki, Hisashi Moriizumi.

**Writing – review & editing:** Toshihito Umegaki, Hisashi Moriizumi, Fumiko Ogushi, Takashi Suzuki.

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
