## [Decision Letter · Decision Letter 0]

10 Apr 2024

Dear umegaki Umegaki,

Thank you very much for submitting your manuscript "Molecular dynamics simulations of a multicellular model with cell-cell interactions and Hippo signaling pathway" for consideration at PLOS Computational Biology.

As with all papers reviewed by the journal, your manuscript was reviewed by members of the editorial board and by an independent reviewer. In light of the review (below this email), we would like to invite the resubmission of a significantly-revised version that takes into account the reviewers' comments.

We cannot make any decision about publication until we have seen the revised manuscript and your response to the reviewers' comments. Your revised manuscript is also likely to be sent to reviewers for further evaluation.

Sincerely,

David van der Spoel

Academic Editor

PLOS Computational Biology

Daniel Beard

Section Editor

PLOS Computational Biology

Reviewer's Responses to Questions

**Comments to the Authors:**

Reviewer #1: See attached.

**Have the authors made all data and (if applicable) computational code underlying the findings in their manuscript fully available?**

Reviewer #1: Yes

PLOS authors have the option to publish the peer review history of their article (what does this mean?). If published, this will include your full peer review and any attached files.

Reviewer #1: No
---

## [Decision Letter · Decision Letter 1]

12 Jul 2024

Dear umegaki Umegaki,

Thank you very much for submitting your manuscript "Molecular dynamics simulations of a multicellular model with cell-cell interactions and Hippo signaling pathway" for consideration at PLOS Computational Biology.

As with all papers reviewed by the journal, your manuscript was reviewed by members of the editorial board and by an independent reviewer. In light of the review (below this email), we would like to invite the resubmission of a significantly-revised version that takes into account the reviewers' comments.

We cannot make any decision about publication until we have seen the revised manuscript and your response to the reviewers' comments. Your revised manuscript is also likely to be sent to reviewers for further evaluation.

Sincerely,

David van der Spoel

Academic Editor

PLOS Computational Biology

Daniel Beard

Section Editor

PLOS Computational Biology

Reviewer's Responses to Questions

**Comments to the Authors:**

Reviewer #1: Overall, this is a much improved version of the manuscript. The main innovation of this work is the combination of classical mechanics to describe cellular motions, along with mass-action kinetics to describe intracellular signaling. The methodology seems novel. That said, the work appears incremental with respect to biology.

I think it would be useful if the authors placed the intracellular dynamics and the inter-cellular dynamics in the context of execution dynamics (e.g. https://www.nature.com/articles/s41540-019-0100-9 or https://journals.plos.org/ploscompbiol/article?id=10.1371/journal.pcbi.1004193) as I think this might be relevant to readers (at the very least mention in the discussion).

Figure 1 is also very confusing. It would be useful if there was a schematic that depicts the overall methodology and then a workflow diagram, similar to that in figure 1 but simplified for readers.

**Have the authors made all data and (if applicable) computational code underlying the findings in their manuscript fully available?**

Reviewer #1: Yes

PLOS authors have the option to publish the peer review history of their article (what does this mean?). If published, this will include your full peer review and any attached files.

Reviewer #1: No
---

## [Editor Report · Decision Letter 2]

2 Oct 2024

Dear umegaki Umegaki,

We are pleased to inform you that your manuscript 'Molecular dynamics simulations of a multicellular model with cell-cell interactions and Hippo signaling pathway' has been provisionally accepted for publication in PLOS Computational Biology.

Best regards,

David van der Spoel

Academic Editor

PLOS Computational Biology

Daniel Beard

Section Editor

PLOS Computational Biology

---

## [Editor Report · Acceptance letter]

24 Oct 2024

PCOMPBIOL-D-23-02102R2 

Molecular dynamics simulations of a multicellular model with cell-cell interactions and Hippo signaling pathway

Dear Dr Umegaki,

I am pleased to inform you that your manuscript has been formally accepted for publication in PLOS Computational Biology. Your manuscript is now with our production department and you will be notified of the publication date in due course.

With kind regards,

Dorothy Lannert
